# Impact of Moderate-To-Vigorous Sports Participation Combined with Resistance Training on Metabolic and Cardiovascular Outcomes among Lean Adolescents: ABCD Growth Study

**DOI:** 10.3390/ijerph20010444

**Published:** 2022-12-27

**Authors:** Ana Elisa von Ah Morano, Wésley Torres, Eduardo Zancheti, Andrea Wigna Pereira de Jesus, Jacqueline Bexiga Urban, Romulo Araújo Fernandes

**Affiliations:** Laboratory of InVestigation in Exercise—LIVE, Department of Physical Education, São Paulo State University (UNESP), Presidente Prudente 19060-900, SP, Brazil

**Keywords:** exercise, risk factors, pediatrics

## Abstract

Background: To investigate the combined impact of being engaged in resistance training (RT) and meeting the physical activity guidelines through sports participation (SP) on cardiovascular and metabolic parameters in lean adolescents. Methods: A longitudinal study, part of the ongoing study entitled “ABCD Growth Study” (Analysis of Behaviors of Children During Growth), assessed data from 64 adolescents (23 from the sport group, 11 from the sport + RT group, and 30 from the control group). Metabolic and cardiovascular outcomes were analyzed as dependent variables. For the independent variables, sports participation and resistance training were considered, and for the covariates, sex, chronological age, body weight, height, and somatic maturation. Results: After 12 months of follow-up, the RT + SP presented improvements in triglycerides (TG) and the SP presented a reduction in LDL-c, TG, and glucose when compared to the control group. Conclusions: Being engaged in RT and SP is a good strategy to improve health in eutrophic adolescents, with a great impact on TG from the lipid profile.

## 1. Introduction

Cardiovascular diseases (CVD) represent the main cause of death in adults worldwide [1]. In fact, the first signs of cardiovascular disease manifest in the first decades of life [2], raising the relevance of unhealthy behaviors adopted during adolescence, such as insufficient physical activity [3].

Sports participation (SP) is a subset of physical activity and constitutes the main manifestation of physical exercise during adolescence, being influenced by sociocultural aspects [4]. Engagement in sports is the most important way for adolescents to meet guidelines for moderate-to-vigorous physical activity (MVPA) [5] and is widely recommended for pediatric groups.

Furthermore, engagement in resistance training (RT) is linked to improved sport performance and muscle strength [6]. Moreover, RT contributes to the maintenance or increase in muscle mass, control of body fatness [7,8], increase in the resting metabolic rate, and, potentially, an increase in daily caloric expenditure [6,9]. Additionally, RT appears to have a direct relationship with increased bone mineral density [10] and improved resting blood pressure, without affecting growth [11].

Although there are solid guidelines recommending both SP and RT for adolescents, their combined effect on cardiovascular and metabolic aspects of adolescents is still under investigation. Pediatric exercise science has extensively investigated the impact of RT and SP on the health aspect of overweight/obese adolescents but in lean adolescents this information is unclear.

Thus, the aim of this manuscript was to elucidate the features of systematic training and physical activity in adolescents with different metabolic characteristics. Additionally, to analyze the combined impact of being engaged in RT and meeting the physical activity guidelines through SP on cardiovascular and metabolic parameters in lean adolescents compared to both only meeting the physical activity guidelines through SP and no engagement in SP or RT at all.

## 2. Materials and Methods

### 2.1. Sampling

This longitudinal study was part of the ongoing study entitled “ABCD Growth Study” (Analysis of Behaviors of Children During Growth), which is being carried out in the city of Presidente Prudente, State of São Paulo, Brazil. The ABCD Growth Study was approved by the Ethics Committee of São Paulo State University (UNESP [process: 1.677.938]). Data collection and analyses were performed by researchers of the Laboratory of Investigation in Exercise (LIVE), which is part of the Department of Physical Education of UNESP. Parents/guardians, and adolescents signed the written consent form.

The ABCD Growth Study is a pragmatic trial in which researchers are interested in the way sports participation affects health outcomes among adolescents in the “real world”, targeting improved external validity of the findings. Therefore, researchers monitor adolescents over time but do not interfere in their training routine, different from a clinical trial where the variables (e.g., training, rest, nutrition, etc.) are controlled in order to reach maximum internal validity (ideal conditions) while reducing external validity.

The dataset for this manuscript was collected in 2017 (baseline) and 2018 (12 months of follow-up). Further details about the sampling process can be found elsewhere [12,13]. Briefly, at baseline, researchers contacted school units and sports clubs spread out across the city to request authorization to contact the adolescents. After receiving permission, the researchers contacted the respective sites and visited them in order to explain the inclusion criteria, which were as follows: (1.) chronological age between 11 and 18 years; (2.) not having any metabolic disorder that hinders sports practice; (3.) at least one year of regular sports practice (sports group); (4.) at least one year without regular practice or exercise (control group); and (5.) signed consent and informed consent forms from legal guardians and adolescents, respectively. Additionally, for this specific manuscript, the absence of obesity was adopted as an inclusion criterion (body fatness < 25% for boys and <30% for girls).

At baseline, 285 adolescents were assessed and started the follow-up period. After 12 months of follow-up, 189 adolescents were reassessed. The reasons for the 96 dropouts were as follows: fear of blood collection, having moved to another city, not having enough time to participate in data collection, and the desire to drop out of the study. From those 189 adolescents who remained in the study, 5 were excluded as they were 18 years old at baseline, 14 were excluded due to missing data in at least 1 of 4 lipid variables at follow-up, 1 adolescent was excluded due to missing data for anthropometry at follow-up, and 63 were excluded due to a diagnosis of obesity at baseline. From the 106 lean adolescents remaining, 42 were excluded due to not meeting the physical activity guidelines through sports participation. Finally, the sample was composed of 64 adolescents (48 boys and 16 girls) (Figure 1).

### 2.2. Dependent Variables: Metabolic and Cardiovascular

Systolic (SBP) and diastolic blood pressure (DBP) were analyzed using an automatic device (Omron brand, Healthcare, model HBP 1100) previously validated for the pediatric population [14]. After the 10 min rest period, three measurements were made in the right arm near the humerus, with a 1 min interval between them, and the average of the three evaluations was adopted. The lipid profile and glycemic profile were evaluated through the following variables: total cholesterol (CT), high-density lipoprotein (HDL-c), low-density lipoprotein (LDL-c), triglycerides (TG), and glucose. The adolescents were instructed to fast for 12 h before the test. 

All variables were treated as absolute changes (Δ) after 12 months of follow-up (subtraction of the baseline values from follow-up values).

### 2.3. Independent Variable: Sports Participation and Resistance Training

Researchers monitored adolescents over a period of 12 months, assisted by coaches and assistant coaches. Information about RT was collected through a face-to-face interview (e.g., days per week, previous time of engagement, etc.), and adolescents who reported RT at baseline and follow-up were considered consistently engaged. The heart rate of adolescents engaged in sports was assessed during two whole training sections (e.g., warm-up, drills, practice itself, cool-down [with a sensor attached to their chest]), and time in moderate-to-vigorous physical activity was calculated (Polar brand, model H7) [15,16]. In this study, only adolescents meeting the physical activity guidelines for sports participation (n = 34; 60 min of moderate-to-vigorous physical activity) were considered. Finally, combining both sports participation and RT, these adolescents were divided according to their engagement in RT into a sport group (n = 23; only sport) and a sport-RT group (n = 11; sport + RT).

Adolescents who were assessed in schools and declared no engagement in sports at baseline (and over the 12 months prior to baseline) were tracked as “control” and only those who remained not engaged in both sports and RT were considered in this manuscript.

### 2.4. Covariates

Sex and chronological age were collected through a face-to-face interview. The sex was defined as the biological sex at birth. Body weight was measured using a scale (Filizola brand, model Personal Line 200) and height and cephalic trunk height using a fixed stadiometer (Sanny brand, Professional model). Body fatness was estimated by bone densitometry (brand General Electric, model WH—Prodigy Primo). Anthropometric data were used to estimate somatic maturation through the peak height velocity (PHV) [17].

### 2.5. Statistical Analyses

Descriptive data are presented as mean, standard deviation (SD), and 95% confidence interval (95% CI). Analysis of variance (ANOVA) and covariance (ANCOVA) were used to compare the variables according to sports participation and RT in crude and adjusted approaches, respectively. Post-hoc tests were used when necessary (Tukey and Bonferroni, respectively). ANCOVA models were adjusted for sex, age, maturation, body fatness, and the baseline values of the dependent variable. Levene’s test assessed the assumption of homogeneity of variances (all models were adequately fit), and measures of effect size were expressed as eta-squared (ES-r) as follows: ES-r < 0.064 (small effect size), ES-r ≥ 0.064 and < 0.140 (moderate effect size), and ES-r ≥ 0.140 (high effect size). Statistical significance was set at 5% (*p*-value < 0.05) and analyses were performed using the software BioEstat (version 5.0).

## 3. Results

### 3.1. Characteristics of the Sample

Baseline characteristics of the sample, stratified according to groups, are presented in Table 1. The sport group was younger than the control group (*p*-value = 0.001) and the sport-RT group (*p*-value = 0.019). Similarly, the PHV of the sport group was lower than the control group (*p*-value = 0.001) and the sport-RT group (*p*-value = 0.001), although adolescents in all groups had passed the peak. The glucose level was lower in the control group than the sport-RT group (*p*-value = 0.007).

### 3.2. Changes in Metabolic and Cardiovascular Outcomes

Table 2 presents the absolute changes after 12 months, divided according to engagement in sports and RT. For RT, there was a difference only for changes in TG (*p*-value = 0.033). In parallel, when the data were divided by SP, there were differences for LDL-c, TG, and glucose (*p*-value = 0.005, *p*-value = 0.001, and *p*-value = 0.007, respectively).

After 12 months, the control group presented a significant increase for LDL-c (6.8 mg/dL [0.45 to 13.1]), TG (16.0 mg/dL [7.0 to 24.9]), and glucose (4.2 mg/dL [0.9 to 7.5]), which was higher than observed for the sport group (LDL-c, TG, and glucose) and sport + RT group (TG) (Table 3).

ANCOVA identified that even after controlling for variance explained by the confounding factors, adolescents who were simultaneously engaged in RT and SP presented lower TG levels (−7.1 mg/dL [95% CI: −22.1 to 7.9]) than the control group (15.3 mg/dL [95% CI: 5.5 to 25.2]). The magnitude of the difference was moderate. Moreover, differences for LDL-c and glucose did not remain significant in the adjusted models, but the effect size observed for LDL-c was of moderate magnitude (Table 4).

## 4. Discussion

This pragmatic trial identified that simultaneous engagement in both sports and RT was significantly related to moderate improvements in TG.

The first aspect of our findings is the fact that, from the seven outcomes investigated, only one was significantly related to the combined engagement in sports and RT. In fact, this is not a surprise, mainly because the adolescents assessed were non-obese, and thus, abnormal values are less common [18]. However, the effect size observed was moderate (even for the outcomes that were not significant [LDL-c]) [19,20], which is surprising because in non-obese adolescents, a small magnitude was expected. Our findings highlight the relevance of sports participation to metabolic health, not just the engagement itself, but the regular engagement in sports enough to meet the physical activity guidelines as a relevant way to promote metabolic benefits even in lean adolescents.

To our knowledge, there are limited data investigating the combined impact of RT and sports participation on cardiovascular and metabolic parameters in lean adolescents. Our main findings regarding the combined impact of both on lipids are in agreement with other studies assessing sports participation [21,22], but include adolescents of different weight categories. The combined engagement in RT and aerobic exercise is highly recommended in guidelines to treat cardiometabolic comorbidities related to pediatric obesity [23,24,25,26], and our findings indicate that this effect is also seen in lean adolescents.

In terms of glucose parameters, differences became non-significant on the adjusted model. The absence of significant differences for glucose was attributed to differences at the baseline (which were higher in the group engaged in both SP and RT), although no abnormal values were observed for glucose in the sample (glucose > 126 mg/dl [27]). Therefore, it seems reasonable to believe the combination of RT and sports participation at a moderate-to-vigorous intensity is beneficial for the glucose metabolism of lean adolescents, even in clinical cases where glucose values are in the normal range. In fact, further studies about the issue are needed, mainly considering larger sample sizes than in the current pragmatic trial.

The background supporting the use of exercise to treat obese adolescents is consistent. Data provided by a randomized clinical trial indicate that the impact of aerobic exercise combined with strength training in overweight and obese adolescents is one of the best strategies to achieve recommended lipid profile values [28]. In addition, aerobic exercise combined with strength training is a very widely recommended non-pharmacologic strategy to prevent and treat diseases such as dyslipidemia and diabetes mellitus type II [27]. The main question raised by us was whether this beneficial impact is observed in the absence of obesity, and our findings seem to confirm its existence. Moreover, the literature points out that the benefits attributed to physical exercise are highly determined by the exercise intensity [29], while our findings also support the relevance of activities of higher intensity in order to achieve these benefits.

The limitations of the present study should be recognized. First, the absence of training parameters for RT (e.g., intensity, frequency, volume, etc.) is relevant because, without these data, it is not possible to describe in depth how RT was administered to these adolescents. Second, the pragmatic approach adopted in this study provides a better inference of these findings in the reality of these adolescents’ lives. In fact, the combination of different sports gives a “real-world” to our findings but limits the possibility of describing which sport would be more beneficial for the metabolic health of lean adolescents, if any. Finally, the reduced sample size, especially in the group SP + RT, limits sex-specific analyses.

## 5. Conclusions

In summary, combined engagement in sports of moderate-to-vigorous intensity and RT seems a relevant strategy to improve lipid profiles in lean adolescents.

## Figures and Tables

**Figure 1 ijerph-20-00444-f001:**
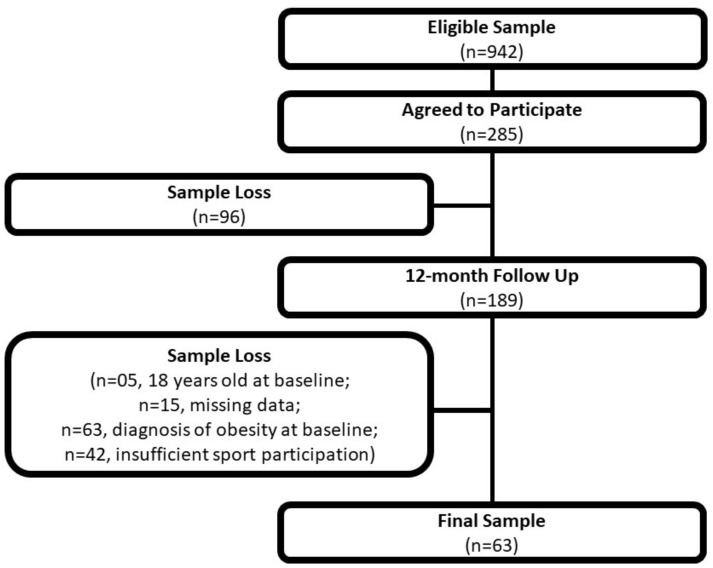
Flow chart of the longitudinal study (Presidente Prudente, Sao Paulo, Brazil, 2017–2018).

**Table 1 ijerph-20-00444-t001:** Baseline characteristics of the adolescents according to their engagement in sports and RT.

	Control (n = 30)	Sport (n = 23)	Sport + RT (n = 11)	ANOVA
	Mean (95% CI)	Mean (95% CI)	Mean (95% CI)	*p*-Value
Boys/Girls	24/06	18/05	06/05	0.254
Age (years)	**16.07** **(15.64 to 16.51) ^a^**	**13.74** **(13.09 to 14.38) ^c^**	**15.07** **(14.24 to 15.90)**	**0.001**
Body weight (kg)	56.13 (52.86 to 59.40)	52.83 (48.62 to 57.05)	54.36 (48.68 to 60.04)	0.428
Height (cm)	170.47 (167.97 to 172.97)	168.68 (163.73 to 173.63)	166.86 (162.84 to 170.88)	0.467
PHV (years)	**1.94** **(1.59 to 2.30) ^a^**	**0.42** **(−0.09** to **0.93) ^c^**	**2.06** **(0.95 to 3.17)**	**0.001**
Body fatness (%)	15.77 (13.04 to 18.51)	14.85 (12.67 to 17.03)	16.50 (12.80 to 20.20)	0.753
LST (kg)	44.1 (41.1to 46.9)	42.1 (38.1 to 45.9)	42.6 (37.4 to 47.9)	0.662
TC (mg/dL)	132.73 (125.04 to 140.43)	139.61 (128.30 to 150.91)	130.27 (117.69 to 142.86)	0.422
HDL-c (mg/dL)	55.23 (49.87 to 60.60)	52.35 (47.49 to 57.20)	54.27 (46.39 to 62.15)	0.721
LDL-c (mg/dL)	62.76 (55.46 to 70.06)	71.82 (61.94 to 81.71)	61.59 (50.13 to 73.04)	0.218
TG (mg/dL)	64.93 (55.91to 73.95)	70.48 (62.49 to 78.47)	63.82 (53.68 to 73.95)	0.560
Glucose (mg/dL)	**80.10** **(77.05 to 83.14) ^b^**	**86.06** **(83.04 to 89.07)**	**97.06** **(74.78 to 119.33)**	**0.009**
SBP (mmHg)	113.67 (110.05 to 117.30)	111.40 (105.47 to 117.33)	112.33 (103.56 to 121.10)	0.785
DBP (mmHg)	**65.08** **(62.44 to 67.72)**	**60.79** **(58.12 to 63.46)**	**60.63** **(55.67 to 65.59)**	**0.046**
Categorical	*n* (%)	*n* (%)	*n* (%)	
HDL-c _(<45 mg/dL)_	03 (10%)	03 (13%)	02 (18.2%)	0.524
TG _(≥90 mg/dL)_	01 (3.3%)	00 (0%)	00 (0%)	0.297
Glucose _(≥100 mg/dL)_	01 (3.3%)	01 (4.3%)	01 (9.1%)	0.555
HBP	03 (10%)	01 (4.3%)	01 (9.1%)	0.618

Notes: Statistical difference between (a.) Control group vs. Sport group, (b.) Control group vs. Sport-RT group, and (c.) Sport group vs. Sport-RT group. PHV = peak high velocity; TC = total cholesterol; HDL-c = high-density cholesterol; LDL-c = low-density cholesterol; TG = triglycerides; SBP = systolic blood pressure; DBP = diastolic blood pressure; HBP = hypertension blood pressure; and LST = lean soft tissue. Statistically significant values (*p* < 0.05) are in bold.

**Table 2 ijerph-20-00444-t002:** Changes in metabolic and cardiovascular outcomes after 12 months of follow-up in adolescents according to engagement in RT and sports.

	RT—Yes (n = 11).	RT—No (n = 53)	
	Mean (SD)	Mean (SD)	*p*-Value
Body fatness (%)	1.40 (1.13)	1.70 (3.77)	0.807
TC (mg/dL)	2.90 (13.04)	0.03 (21.48)	0.664
HDL-c (mg/dL)	1.45 (8.14)	−1.39 (8.96)	0.334
LDL-c (mg/dL)	2.14 (11.24)	0.51 (17.39)	0.768
TG (mg/dL)	**−5.36 (15.14)**	**7.77 (26.31)**	**0.033**
Glucose (mg/dL)	−14.64 (31.26)	1.41 (8.32)	0.121
SBP (mmHg)	1.87 (9.66)	3.00 (9.39)	0.721
DBP (mmHg)	2.33 (94.91)	1.12 (6.51)	0.564
	**Control (n = 30)**	**Sport (n = 34)**	
	**Mean (SD)**	**Mean (SD)**	***p*-Value**
Body fatness (%)	2.36 (3.24)	1.03 (4.08)	0.159
TC (mg/dL)	5.50 (21.47)	−3.97 (18.23)	0.061
HDL-c (mg/dL)	−3.10 (8.47)	1.02 (8.81)	0.061
LDL-c (mg/dL)	**6.80 (16.99)**	**−4.50 (14.15)**	**0.005**
TG (mg/dL)	**16.03 (23.95)**	**−3.76 (22.69)**	**0.001**
Glucose (mg/dL)	**4.25 (8.95)**	**−6.27 (18.77)**	**0.007**
SBP (mmHg)	5.03 (9.27)	0.84 (9.15)	0.074
DBP (mmHg)	0.83 (6.17)	1.77 (6.36)	0.552

Notes: TC = total cholesterol; HDL-c = high-density cholesterol; LDL-c = low-density cholesterol; TG = triglycerides; SBP = systolic blood pressure; and DBP = diastolic blood pressure. Statistically significant values (*p* < 0.05) are in bold.

**Table 3 ijerph-20-00444-t003:** Crude changes in metabolic and cardiovascular outcomes after 12 months of follow-up in adolescents according to the combined engagement in sports and RT.

	Control (n = 30)	Sport (n = 23)	Sport + RT (n = 11)	ANOVA
	Mean (95% CI)	Mean (95% CI)	Mean (95% CI)	*p*-Value
TC (mg/dL)	5.50 (−2.52 to 13.52)	−7.26 (−15.76 to 1.24)	2.90 (−5.85 to 11.67)	0.066
HDL-c (mg/dL)	−3.10 (−6.26 to 0.06)	0.82 (−3.18 to 4.84)	1.45 (−4.01to 6.92)	0.173
LDL-c (mg/dL)	**6.80** **(0.45 to 13.14) ^a^**	**−7.67** **(−13.95 to −1.40)**	**2.14** **(−5.41 to 9.70)**	**0.005**
TG (mg/dL)	**16.03** **(7.09 to 24.97) ^a,b^**	**−3.00** **(−14.16 to 8.16)**	**−5.36** **(−15.53 to 4.80)**	**0.005**
Glucose (mg/dL)	**4.25** **(0.90 to7.59) ^a^**	**−2.27** **(−4.75 to 0.20)**	**−14.64** **(−35.64 to 6.35)**	**0.002**
SBP (mmHg)	5.03 (1.57 to 8.49)	0.34 (−3.57 to 4.27)	1.87 (−4.61to 8.37)	0.186
DBP (mmHg)	0.83 (−1.47 to 3.14)	1.50 (−1.53 to 4.55)	2.33 (−0.22 to 2.89)	0.787

Notes: Statistical difference between (a.) Control group vs. Sport group, and (b.) Control group vs. Sport-RT group. TC = total cholesterol; HDL-c = high-density cholesterol; LDL-c = low-density cholesterol; TG = triglycerides; SBP = systolic blood pressure; and DBP = diastolic blood pressure. Statistically significant values (*p* < 0.05) are given in bold.

**Table 4 ijerph-20-00444-t004:** Adjusted absolute changes in metabolic and cardiovascular outcomes after 12 months of follow-up in adolescents according to the combined engagement in sports and RT.

	Control (n = 30)	Sport (n = 23)	Sport + RT (n = 11)	ANCOVA
	Mean (95% CI)	Mean (95% CI)	Mean (95% CI)	*p*-Value ^effect-size^
LDL-c (mg/dL)	5.60 (0.08 to 11.12)	−3.95 (−10.55 to 2.64)	−2.36 (−10.87 to 6.13)	0.104 ^0.079 (Moderate)^
TG (mg/dL)	**15.38** **(5.51 to 25.25) ^a^**	**−1.30** **(−13.08 to 10.47)**	**−7.13** **(−22.21 to 7.94)**	**0.033 ^0.117 (Moderate)^ ***
Glucose (mg/dL)	0.455 (−2.06 to 2.91)	−2.76 (−5.68 to 0.14)	−3.27 (−7.30 to 0.76)	0.212 ^0.055 (Small)^

Notes: Statistical difference between (a.) Control group vs. Sport + RT group. RT = resistance training; 95% CI = 95% confidence interval; ANCOVA = analysis of covariance. LDL-c/model with covariates: sex (*p*-value = 0.219), age (*p*-value = 0.412), maturation (*p*-value = 0.263), body fatness-baseline (*p*-value = 0.538), body fatness-change (*p*-value = 0.299), and *TG (*p*-value = 0.001). Statistically significant values (*p* < 0.05) are in bold.

## Data Availability

The data presented in this study are available on request from the corresponding author. The data are not publicly available due to privacy.

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
