# Peer review of "Impact of Moderate-To-Vigorous Sports Participation Combined with Resistance Training on Metabolic and Cardiovascular Outcomes among Lean Adolescents: ABCD Growth Study"

_ijerph, 2022, doi:10.3390/ijerph20010444_

Round 1

Reviewer 1 Report

Dear editor and authors,

Thank you for the opportunity to review this manuscript. In this article, the authors use data from a longitudinal study on lean adolescents to evaluate how sports participation (SP) and resistance training (RT) influence cardiometabolic outcomes. Participants were evaluated at baseline and after a one-year period, and they were classified into SP, SP + RT, and control groups. The authors observed beneficial effects of SP and RT on cardiometabolic variables; after controlling for confounding factors, the SP + RT group had moderately lower triglyceride levels than the control group. This study is novel because it focuses on effects of physical activity in lean adolescents rather than those with overweight or obesity. Other strengths of the study include its longitudinal design as well as the use of bone densitometry to measure body fatness. Some weaknesses are the small number of female participants (both in absolute terms and as a proportion of all participants), evaluating RT participation at only two time points, and measuring physical activity levels at only two training sessions. Overall, this is a well-written article that strengthens the literature on the key role of physical activity in youth to reduce cardiovascular disease risk. Please see my comments below for revisions to make and questions to address before the manuscript is ready for publication.

Comments:

- The introduction and discussion are concise. The introduction logically lays out the rationale for the study, while the discussion situates the findings in the context of the existing body of literature on this topic.

- Please do not begin sentences with an abbreviation (e.g., in the introduction, “RT contributes to…”).

- Please write out numbers smaller than 10 (e.g., “one” instead of “1”).

- How was the gender/sex of the participants determined? Was their biological sex ascertained, or did the authors categorize them as “boys” and “girls” based on their gender identity? In the methods section, please clarify whether “boys” and “girls” refers to gender identity or is linked to the sex assigned at birth of the participants.

- After breaking down the participants into the control group, SP-only group, and SP + RT group, the sample sizes for SP + RT was fairly small. Was the study adequately powered to detect differences between the groups?

- It is my understanding that Spearman correlation is not an appropriate test to use for correlating a yes/no variable with a quantitative variable (Table 3). Additionally, the authors do not mention the use of Spearman correlation to evaluate relationships between MVPA and cardiometabolic variables.

- In the results section, in the text referring to the data presented in Table 4, please cite Table 4 so that readers know where to find that information. Also regarding Table 4: Did the authors do post-hoc tests to determine which group(s) were significantly different from the others when an overall difference was detected? Please note this in the methods section.

- In the results, effect size is mentioned only once. It might be valuable to report other effect sizes.

- Throughout the paper, please be mindful of the use of “decrease” or “increase” vs. “lower” or “higher.” It is sometimes difficult to determine whether a cited statistical difference is between groups for the magnitude change in a variable from baseline to follow-up, or between groups for an absolute level of a variable.

- The first sentence of the discussion does not make grammatical sense.

- In the discussion, the authors state that there were no significant findings regarding glucose. Table 4 reports that the crude changes in glucose were significantly different between controls and SP or SP + RT groups. The authors also state that the effect size was great, but effect size is not discussed elsewhere in the paper. As noted in my previous comments, please include more data on effect sizes in the results section.

Author Response

Ref: Manuscript ID: ijerph-2060146

Impact of moderate-to-vigorous sports participation combined with resistance training on metabolic and cardiovascular outcomes among lean adolescents

We are pleased to resubmit the above manuscript which required major revision. We thank the reviewers for their kindness and critiques which have enabled us to improve the manuscript considerably. We have addressed all the points raised by the reviewers and trust that the manuscript is now suitable for publication. For the changes, we used the "Track Changes" function in the text manuscript and all of them are detailed point by point on response document.

Respectfully,

Rômulo Araújo Fernandes, PhD

Laboratory of InVestigation in Exercise – LIVE

Universidade Estadual Paulista (UNESP)

Reviewer 2 Report

I have read the manuscript with interest, and I must admit that I am no expert in metabolic and cardiovascular outcomes. Most parts of the manuscript and study performed were clear to me though. I have written some comments on parts that were not, or needed more explanation in my opinion. 

Abstract:

According to the methods in your abstract there is no sport+RT group? But in the results this group is mentioned. 

Introduction: 

Add the comparison in the aim: "the main aim of this manuscript was to analyze the combined impact of being engaged in RT and meeting the physical activity guidelines through SP on cardiovascular and metabolic parameters in lean adolescents, compared to.......

Methods and materials:

If possible regarding the number of figures/tables used, please add a flow chart for your study sample, and the groups (sports only) (sport+RT) (control) and numbers you included in your analyses.  

Additionally. In the abstract you mention that data from 53 adolescents were assessed, but at the end of the sampling paragraph, you mention that the sample was composed of 64 adolescents. Where are the other 11? I guess this is the sport+RT group?

Only at the end of the paragraph independent variables it becomes clear what your comparison is. Please, mention your comparison in the aim.

Controls are considered as controls if they remained not engaged in sports and RT for 12 months. Does this account for the Sports only and the sport+Rt group as well? What happened with those participants who started to participate in RT during the 12 months follow-up period?

Results section:

Table 1: Please check superscript a en c in Age. The 95%CI of sport and sport+Rt are overlapping. Is the p-value 0.019 correct?

The same goes for the 95% CI for glucose in the control group and sport group. Are the intervals or is the p-value correct?

What is shown in 3? I don't understand this one. 

Please check intervals TG in table 4. Are they correct? The lower bound in sport and sport+RT is higher than the upper bound. 

The 95%CI in sport+RT for Glucose is very wide and almost completely the 95%CI of the controls. Is this correct?

Discussion section: please check first sentence.

You mention that your findings highlight the relevance of sports participation to metabolic health, not just the engagement itself, but the regular engagement in sports to meet the physical activity guidelines as a relevant way to promote metabolic benefits even in lean adolescents. 

The TG levels were already good in all three groups < 150 mg/dL. How beneficials is the decrease in TG than? And, can de decrease be caused by differences/changes in nutrition patterns of water intake?

I absolutely believe that being physically active in sports and RT is beneficial for one's health, but I cannot judge how valuable the results of your study are, as this is not my expertise.

Minor revisions:

- I believe there is a space too much after engaged in.. Fist sentence in the abstract.

- Check the following sentence just above the aim in the introduction section for correctness: The absence of information about the impact of RT and SP on health aspects seems even greater in lean adolescents, since although Pediatric Exercise Science as a discipline has extensively investigated this phenomenon among obese adolescents, limited information is provided for those not affected by obesity.  

Author Response

(The authors gave the same response as above.)

Reviewer 3 Report

In my opinion, the effect on lipid status in adolescents of different groups has not been fully proven. The article is interesting and gives new information, especially about the metabolism of lean adolescents. But there are some questions. Perhaps it is possible to limit the purpose of the work to the features of systematic training and physical activity of adolescents with different metabolic characteristics.

Author Response

(The authors gave the same response as above.)

Round 2

Reviewer 2 Report

Dear authours,

Thank you for the point by point reply. I have just one minor comment left. Your final sample is 64 I believe, and not 63 > see figure 1.